# The Role of Hydrogen Incorporation into Amorphous Carbon Films in the Change of the Secondary Electron Yield

**DOI:** 10.3390/ijms241612999

**Published:** 2023-08-20

**Authors:** Nenad Bundaleski, Carolina F. Adame, Eduardo Alves, Nuno P. Barradas, Maria F. Cerqueira, Jonas Deuermeier, Yorick Delaup, Ana M. Ferraria, Isabel M. M. Ferreira, Holger Neupert, Marcel Himmerlich, Ana Maria M. B. do Rego, Martino Rimoldi, Orlando M. N. D. Teodoro, Mikhail Vasilevskiy, Pedro Costa Pinto

**Affiliations:** 1Centro de Física e Investigação Tecnologica, Departamento de Física, Faculdade de Ciências e Tecnologia, Universidade Nova de Lisboa, Campus de Caparica, 2829-516 Caparica, Portugal; c.adame@campus.fct.unl.pt (C.F.A.); odt@fct.unl.pt (O.M.N.D.T.); 2Departamento de Engenharia e Ciências Nucleares, Instituto Superior Técnico, University of Lisbon, 2695-066 Bobadela, Portugal; ealves@ctn.tecnico.ulisboa.pt (E.A.); nunoni@ctn.tecnico.ulisboa.pt (N.P.B.); 3Instituto de Plasmas e Fusão Nuclear, Instituto Superior Técnico, University of Lisbon, 1049-001 Lisbon, Portugal; 4Centre of Physics of the Universities of Minho and Porto (CF-UM-UP) and Laboratorio de Física para Materiais e Tecnologias Emergentes (LaPMET), 4710-057 Braga, Portugal; fcerqueira@fisica.uminho.pt (M.F.C.); mikhail@fisica.uminho.pt (M.V.); 5International Iberian Nanotechnology Laboratory (INL), Av. Mestre José Veiga, 4715-330 Braga, Portugal; 6Centro de Investigação de Materias (Lab. Associade I3N), Departamento de Ciência dos Materiais, Faculdade de Ciências e Tecnologia, Universidade Nova de Lisboa, Campus de Caparica, 2829-516 Caparica, Portugal; j.deuermeier@fct.unl.pt (J.D.); imf@fct.unl.pt (I.M.M.F.); 7European Organization for Nuclear Research, CERN, 1211 Geneva, Switzerland; yorick.maxence.delaup@cern.ch (Y.D.); holger.neupert@cern.ch (H.N.); marcel.himmerlich@cern.ch (M.H.); martino.rimoldi@cern.ch (M.R.); 8Associate Laboratory i4HB—Institute for Health and Bioeconomy at Instituto Superior Técnico, Universidade de Lisboa, Av. Rovisco Pais, 1049-001 Lisbon, Portugal; ana.ferraria@tecnico.ulisboa.pt (A.M.F.); amrego@ist.utl.pt (A.M.M.B.d.R.); 9iBB—Institute for Bioengineering and Biosciences and Departamento de Engenharia Química, Instituto Superior Técnico, Universidade de Lisboa, Av. Rovisco Pais, 1049-001 Lisbon, Portugal

**Keywords:** amorphous carbon, particle accelerators, SEY, XPS, Raman spectroscopy

## Abstract

Over the last few years, there has been increasing interest in the use of amorphous carbon thin films with low secondary electron yield (SEY) to mitigate electron multipacting in particle accelerators and RF devices. Previous works found that the SEY increases with the amount of incorporated hydrogen and correlates with the Tauc gap. In this work, we analyse films produced by magnetron sputtering with different contents of hydrogen and deuterium incorporated via the target poisoning and sputtering of C_x_D_y_ molecules. XPS was implemented to estimate the phase composition of the films. The maximal SEY was found to decrease linearly with the fraction of the graphitic phase in the films. These results are supported by Raman scattering and UPS measurements. The graphitic phase decreases almost linearly for hydrogen and deuterium concentrations between 12% and 46% (at.), but abruptly decreases when the concentration reaches 53%. This vanishing of the graphitic phase is accompanied by a strong increase of SEY and the Tauc gap. These results suggest that the SEY is not dictated directly by the concentration of H/D, but by the fraction of the graphitic phase in the film. The results are supported by an original model used to calculate the SEY of films consisting of a mixture of graphitic and polymeric phases.

## 1. Introduction

The emission of secondary electrons from surfaces plays a key role in the generation of electron multipacting in high-power radio frequency devices and particle accelerators with positively charged beams. In particle accelerators, it can result in the formation of clouds of electrons, causing beam instabilities, deterioration of the vacuum, or heat loads to cryogenics parts of the system, limiting the overall efficiency of the accelerator (i.e., the beam luminosity) [1,2,3,4,5,6]. A possible cure to this problem is to reduce the electron emission by coating the internal surfaces of the vacuum chambers with a thin film of low Secondary Electron Yield (SEY) material. Amorphous carbon (a-C) coatings have been successfully used in the Super Proton Synchrotron at CERN (European Organisation for Nuclear Research) to mitigate the electron multipacting [7], and it is now in the baseline for the High Luminosity upgrade of the Large Hadron Collider, (CERN, Geneva, Switzerland) and the Electron Ion Collider, (Brookhaven National Laboratory, Upton, USA) [8,9].

Secondary electron emission can be conveniently described as follows [10]. Primary electrons that penetrate a material mainly lose their energy through different types of electron excitations (via plasmon excitations or by a direct energy transfer to valence and core electrons). This results in the generation of secondary electrons inside the material, with energy above the vacuum level. On their way out, these internal secondary electrons can efficiently lose their energy only through the excitation of valence electrons. Finally, once they reach the surface, only the internal secondary electrons which still have energy above the vacuum level have a chance of being emitted. The whole process is strongly affected by the electronic structure of the material, which determines the energy loss processes of both primary and internal secondary electrons. Particularly important is the latter, which can be reduced by opening the energy gap. This is the origin of the high SEY of dielectric materials [11].

The efficiency of a-C coatings for suppressing electron multipacting depends on the structural properties: only coatings with pronounced electrical conductivity can have a low SEY. It has been well-established that the presence of hydrogen impurities is particularly harmful in that respect [12,13]. The effects of hydrogen and nitrogen impurities on the secondary electron emission properties of a-C coatings were the subject of two recent studies [14,15]. It was clearly demonstrated that adding hydrogen or deuterium in the discharge gas increases the SEY of carbon films, while the presence of nitrogen allows for the opposite, which can be used to compensate the effect of hydrogen. By combining SEY measurements with Ion Beam Analysis (IBA), a correlation between the deuterium content and the maximum SEY (SEY_max_) was established. The SEY_max_ increases linearly, from 1 to 1.4, when the overall hydrogen and deuterium (H + D) relative content increases up to 47%. A further increase of the H + D amount up to 54% is accompanied by a steep SEY_max_ growth to 2.2. The samples were also investigated using Optical Absorption Spectroscopy (OAS), enabling the estimation of the optical energy gap using the procedure established by Tauc (the so-called Tauc gap). These results reveal a strong correlation between the SEY_max_ and the Tauc gap, apparently offering a straightforward explanation for the SEY_max_ increase. However, a-C films are often non-uniform, i.e., they may consist of various regions with different compositions and electronic structures (e.g., graphitic, diamond-like, and hydrocarbon domains—the latter being a result of highly common hydrogen contamination) [16]. Such materials have different local energy gaps, limiting the interpretation of Tauc plots [17].

Robertson summarised the measurements of the Tauc gap performed on different types of a–C (pure and hydrogenated) in his seminal review paper [18]. He concluded that the Tauc gap of a film is not determined by the hydrogen content, but by the relative amount and properties of the sp^2^ carbon phase. The local energy gaps of diamond-like and various hydrocarbon phases are too large to be related with a Tauc gap below ≈ 2 eV. The gap is therefore related to the configuration of π states on the sp^2^ sites. In a planar cluster model, the band gap of a given cluster is inversely proportional to the square root of the number of the hexagonal rings in the cluster [16]. Different types of defects (e.g., formation of pentagons or heptagons) will also open the gap. From this perspective, deviation from the linear dependence at low photon energies in Tauc plots should not be attributed to the so-called tail states (i.e., Urbach tail), but to the small quantity of graphitic clusters with small local energy gaps.

The abrupt increase of the SEY_max_, when the H + D relative concentration changes from 47% to 54%, suggests that the hydrogen content may not be directly responsible for the change in the secondary electron emission (SEE) properties of a-C films. This doubt is further supported by the high correlation between the SEY_max_ and the Tauc gap [14,15], with the knowledge that the Tauc gap can be independent of the hydrogen concentration [18]. This aspect motivates our study: as the Tauc gap in a-C is known to be affected by the concentration and size of the graphitic domains, could the latter also influence the SEE?

In our recent work, we performed the deposition of a-C coatings using magnetron sputtering in Ar discharges with several fractions of D_2_ and identified the mechanism behind the incorporation of deuterium in the films during the production phase and its impact on the SEY [15]. It is expected that the incorporation of deuterium affects the electronic structure of a–C, and therefore its SEY, in practically the same way as hydrogen impurities would. At the same time, adding deuterium allows these intentionally added species to be distinguished from the natural hydrogen contamination that originates from the residual background of H containing molecules in the coating system during film deposition, as well as from the surface contamination that mainly derives from hydrocarbons after removal from the vacuum system. In this work, we provide detailed analysis of the same samples by means of different electron and vibrational spectroscopic techniques. X-ray and UV Photoelectron Spectroscopies (XPS and UPS, respectively) were used to determine the surface composition, to identify different phases in the a-C films, and to obtain information on their electronic structure. Further insights are provided using Raman scattering, Fourier Transform Infrared Spectroscopy (FTIR), and High-Resolution Electron Energy Loss Spectroscopy (HREELS). The Raman scattering characterisation, which enabled the detection of the presence of graphitic carbon, was particularly valuable in supporting the interpretation of the XPS results. FTIR and HREELS were used to identify the character of the C–D bonds in the most contaminated samples. These results are then compared with the corresponding SEY, OAS, and IBA measurements, enabling us to reveal the actual mechanism behind the SEY increase caused by the hydrogen contamination.

## 2. Results

### 2.1. Composition, Electron Emission and Optical Properties of the Coatings

In this section, we summarise the main results of the thin films obtained using different techniques and reported in [15]. Thin film composition analysis, performed by IBA, revealed the presence of uniformly distributed C, D, H, and O through the whole film depth, as confirmed by the Secondary Ion Mass Spectrometry measurements. The results of the quantitative composition, the mass densities of the films—estimated from the IBA measurements—and the film thicknesses (obtained by Scanning Electron Microscopy), as well as the results of the SEY and Tauc gap (*E_T_*) measurements, are summarised in Table 1. The amount of incorporated deuterium steadily increases for an increasing D_2_ partial pressure in the discharge gas (*p_D2_*), getting relatively close to the theoretical maximum of 65 at.% in saturated hydrocarbons. The dependence of the maximum secondary electron yield (*SEY_max_*) on *p_D2_* in the films has a different trend: an increase of the H/D amount from 10 to 46 at.% changes *SEY_max_* from 0.99 to 1.38, while additional 7 at.% of H/D increases the *SEY_max_* abruptly to 2.2. Furthermore, the SEY_max_ and the Tauc gap are strongly correlated. It appears that small changes of the D/H content in samples 1D and 10D are accompanied by very different electronic structures, which govern both the secondary electron emission and optical properties. The latter is also clear from the fact that the sample 1D was conductive, in contrast to the sample 10D, which appeared to be insulating. This aspect required the measurement of the SEY of this charging sample in a different experimental setup, using a pulsed electron beam [15].

Based on the mass densities and the Tauc gaps of the films, we can already make a first estimation of their structure using the general guidelines summarised in [19] and by considering that deuterated carbon films are generally somewhat denser than their hydrogenated equivalent. It appears that the reference sample, as well as the 0.2D and 0.5D samples, with their Tauc gaps being clearly below 1 eV, are most likely dominated by graphitic and partially deuterated graphite-like (a-C:D) regions. The 1D sample, which exhibits a high density, a large H + D content leading to *E_T_* = 1.38 eV, high density, and a large H + D content, might be closest to diamond-like a-C:D. Finally, the 10D sample, characterised by a low density and *E_T_* > 2 eV, could be a typical example of a polymer-like a–C:D.

### 2.2. XPS Measurements

Apart from carbon, the survey XPS spectra of all the samples show the presence of ~8 at.% of oxygen at the surface. The samples were not charging during the measurements, except for the sample 10D, which was therefore measured utilising an electron flood gun for charge compensation. The O 1s line in the first four samples was situated at 532.6 eV. Assuming that the O 1s line position is the same in the sample 10D, the latter was used as a binding energy reference for that sample, and all the spectra were shifted accordingly.

Hydrogenated/deuterated a-C is expected to be a mixture of graphitic (pure sp^2^ hybridised) carbon, diamond-like carbon and different hydrocarbons, in which carbon could be in both sp^2^ and sp^3^ hybridisation. Resolving these distributions in the C 1s photoelectron line is a complex problem due to their quite small chemical peak shifts, as extensively discussed in the literature [20,21,22,23]. The graphitic contribution is located at 284.3–284.5 eV, while a pure sp^3^ C contribution is usually encountered at about 285.2 eV [21]. The latter feature overlaps with different hydrocarbon contributions, around 285.0 eV, while at higher binding energies, further contributions related to various carbon bonds with O can be expected [23,24]. All peaks should be symmetric apart the one from the pure graphitic contribution, which is highly asymmetric [22,23]. The peak model for the graphitic contribution could be established by measuring the C 1s photoelectron line of a freshly cleaved Highly Oriented Pyrolytic Graphite (HOPG) crystal. Unfortunately, the asymmetry of the graphitic contribution becomes less pronounced with the cluster size reduction and with the increase in the amount of defects in this phase. These factors make bond identification through fitting the C 1s line spectrum a challenging task, without a straightforward solution. However, the elemental O concentration can be used as an additional constraint, which the carbon-oxygen bond contributions in the C 1s line need to follow. This approach, applied to all the samples except 10D (see below), strengthens the reliability of the interpretation of the spectra taken from the a-C coatings.

The high-resolution spectra of the C 1s photoelectron line of a freshly cleaved HOPG and the five samples considered in this study, together with the corresponding fittings, are presented in Figure 1. In all cases the background was calculated using the Shirley algorithm. As in [14], we again observe that the C 1s line becomes more symmetric with the increase of the D content in the films. One possible explanation of this trend is that the relative amount of graphitic C is reduced and/or that the clusters become smaller with the increase of the D content.

The C 1s spectrum of HOPG (Figure 1a) was fitted using three contributions. The main peak (graphite) at 284.45 eV has an asymmetric profile with a nomenclature LA(1.2, 2.5, 5) in the CasaXPS software [25] and a full width at half maximum (FWHM) of 0.34 eV. It should be stressed that the same peak profile was used by M. Biesinger, and failed to secure perfect agreement with the measured spectrum [23]. As a similar problem appeared in our spectra, we introduced a second peak (“graphite sat”), with the same profile and a FWHM of 1.3 eV. This peak is shifted with respect to the first towards higher binding energy by 1.47 eV, while their intensity ratio is 0.106. Finally, there is a well-known wide symmetric peak at about 291.2 eV attributed to the π-π* satellite always present in spectra of pure graphite [22,23].

It might be tempting to interpret the peak “graphite sat” as evidence of defects, such as sp^3^ carbon or C-H bonds. However, one should bear in mind that such contributions would be symmetric. The asymmetry of this contribution would then imply the non-negligible presence of other contaminants (typically O). that XPS is distinctively more sensitive to such contaminants than to C [26], the corresponding photoelectron lines would be clearly observable. Since this is not the case, we assume that both contributions originate from perfect graphite. They are introduced to better describe the shape of the C 1s line of HOPG, without attributing any physical interpretation to this separation. In our fittings of the graphitic contribution, we fixed the relative positions of the first two peaks and kept their intensity ratio approximately constant (see below). The intensity of the third peak was considered to be a free parameter (the latter is affected by the electronic structure and size of the graphitic clusters). The profile of all the symmetric peaks (attributed to less conductive phases in a-C films) was of pseudo-Voigt type GL(30). They are attributed to sp^3^ carbon and C-H(D) bonds with both sp^2^ and sp^3^ carbon (284.8–285.2 eV), C-O (≈286.6 eV), C=O (287.8–288.2 eV), and O-C=O (≈289.0 eV) bonds [21,24]. All symmetric peaks within one spectrum have FWHMs fixed to the same value.

The essential element of the fitting procedure was to maintain the intensities of the peaks attributed to the carbon-oxygen bonds in the C 1s line in accordance with the O:C concentration ratio obtained from the quantitative analysis, being 8 ± 0.8% for all samples. In this approach, we consider that the C:O stoichiometric ratio is 1:1 for the C-O and C=O contributions and 2:1 for the COO contribution. Consistency between the composition and the C 1s fits could only be achieved by successively decreasing the asymmetry of the graphitic line as the H + D amount increases. The degree of asymmetry is characterised here by the ratio of the parameters α and β of the line shape LA(α,β,m), which determine the spread of the tail on the high and low binding energy side, respectively (α/β being below 1 for asymmetric line shapes). In addition, final asymmetry tuning of the graphitic contribution was performed through a slight modification of the intensity ratio between the main peak and the “graphite sat” in the range 0.078–0.106. This peak fitting model successfully describes the spectra of almost all of the samples; however, it did not provide consistent results for the sample 10D, as can be seen in Table 2, in which the XPS fitting results are summarised. For this sample, the oxygen content is overestimated, most likely due to the charging problems that affected the shape of the C 1s line.

The fitting results of the C 1s line taken from the five samples and the relative content of hydrogen and deuterium taken from Table 1 are summarised in Table 2. The following general trends with the increase of the H + D relative content appear:The relative contribution of the graphitic component decreases and even disappears in the sample 10D;The relative contribution attributed to the sp^3^ carbon and the hydrocarbons increases;The asymmetry of the graphitic component becomes less pronounced.

Therefore, increasing the amount of incorporated H/D in the films is accompanied by a higher content of hydrocarbons and a reduction in the graphitic content. Besides, the graphitic regions become smaller and/or with higher defect concentration.

At this point, the total disappearance of the graphitic content in the sample 10D is not certain. We can only claim that there is no need to introduce the asymmetric graphitic component to obtain a good fit of the spectrum presented in Figure 1f. As the sample 10D was charging, one could expect a deformation of the line shape. Moreover, the latter certainly took place as it was not possible to perform a good fit that would justify the correct content of oxygen. While one cannot reliably exclude the presence of some graphitic regions in that sample, their relative amount most likely does not exceed ≈ 10%.

### 2.3. UPS Measurements

In Figure 2, we present UPS spectra close to the Fermi level (binding energy of 0 eV) of freshly cleaved HOPG, the reference sample without D, as well as of the samples 0.2D, 0.5D and 1D. The spectra are normalised with respect to their intensities at 2 eV. The same structure cannot be reliably measured for the sample 10D due to the charging problems.

The very top of the valence band of the HOPG (binding energy range 0–1 eV) is attributed to π electrons from the K point of the Brillouin zone [27]. As the diamond-like and hydrocarbon regions have energy gaps well above 2 eV [18], the presence of the signal in other samples is additional evidence of the existence of graphitic regions in the a–C samples. The intensity of this structure gradually decreases with the increasing H + D content, as previously concluded from the XPS spectra. A closer inspection of the spectra reveals that the top of the valence band shifts towards a higher binding energy as the H + D content increases. This verifies an opening of the local energy gap, which implies a size reduction of the graphitic regions [16,28] in the films.

The difference between the top of the valence band and the Fermi level (the latter corresponds to the 0 eV binding energy) in the spectra presented in Figure 2 is in the 0.0–0.2 eV range. Assuming that the Fermi level is around the mid-gap position, the energy gap of the graphitic regions can be roughly estimated to be ≈0.2 eV for the reference sample and the 0.2D sample, and ≈0.4 eV for the samples 0.5D and 1D. In a first approximation, the UPS value for the reference sample corresponds to its Tauc gap (Table 1), while the mismatches between the UPS and Tauc values for the other samples (particularly for 1D) are much higher. The latter is due to the character of the Tauc gap, which corresponds to the electronic properties of the dominant components in the film, whereas the UPS detects the highest occupied states. Therefore, reasonable agreement between the Tauc gap measurements and the UPS results of the reference and 0.2D samples is additional evidence of their graphitic character. On the other hand, the mismatch in the case of the other two samples confirms a lower graphitic content, as already revealed by the XPS results.

### 2.4. Raman Spectroscopy

Further insight into the structure of the thin films was obtained using Raman scattering spectroscopy. The results of the Raman spectroscopy performed on the samples deposited on silicon using the 532 nm laser beam are summarised in Figure 3. The representative spectra of most of the samples (excluding the sample 10D) are presented in Figure 3a. The steady increase of the Raman signal with the increasing D_2_ content in the discharge gas is most likely due to a drop of the optical absorption [15], which is also manifested in the increase of the Tauc gap (cf. Table 1), combined with the monotonic increase of the film thickness with the H + D content.

The feature in the 1100–1700 cm^−1^ range corresponds to the overlapping G and D bands, being typical for various carbon phases [29] (including deuterated a–C films [30]). The G band is attributed to sp^2^ carbon atoms (it is not restricted to sixfold carbon rings in graphite nor to pure carbon phases). The D band is a fingerprint of isolated sixfold carbon aromatic rings, being forbidden in a perfect graphite [16]. The co-existence of both bands therefore implies the presence of graphitic regions that are separated by disordered phases.

In addition to the G and D bands, an intense wide structure at about 2800 cm^−1^ can be also observed in Figure 3a. Its intensity relative to the G and D bands increases with the amount of deuterium incorporated in the samples. In the sample 10D, the intensity of this structure is about 10 times greater than in the other samples when measured using the same experimental conditions, thus hindering an identification of the graphitic carbon in this case (more details on this matter are provided in Appendix A). Raman spectra of carbon-based materials may have some bands in this range. For instance, graphene is characterised by the 2D mode at around 2700 cm^−1^ and a second order D + D’ band at around 2900 cm^−1^ [31]; Raman spectra of polymeric amorphous carbon exhibit a C-H stretching band in this range [29,32]. To clarify the origin of this band, Raman measurements were also performed using a He-Ne laser (wavelength 632.8 nm), resulting in a lack of this structure (cf. Appendix A). The absolute energy position of this feature is ≈1.97 eV, corresponding well to the position of the broad emission band (1.9–2.5 eV) observed in a photoluminescence of hydrogenated a–C [33]. Therefore, we conclude that this feature is not a Raman signal, but is evidence of photoluminescence—it is related to the excitation of electronic states. The latter also explains why this feature increases with the increasing deuterium content.

The fitting of the Raman spectra is demonstrated in Figure 3b on the example of the sample 1D, while the other fitting details can be found in Appendix A. The spectra were fitted using four Lorentzian peaks after removing a linear background. Although this approach did not provide a perfect agreement outside the range of interest (the latter being 1100–3000 cm^−1^), it allows the extraction of the most valuable information characterising the a-C samples. Two of the Lorentzians are attributed to the G and D bands at about 1550 cm^−1^ and 1360 cm^−1^, respectively. The peak widths and the position of the D band do not change significantly among the studied samples. There is, however, a substantial difference between the relative peak intensities from the different samples, as well as a clear shift of the G band. The wide hump related to the photoluminescence (PL), was fitted using two contributions, at about 2800 and 2150 cm^−1^. The latter could also be, at least partially, attributed to the CD_x_ stretching band [29] (PL2/C-D); the measurements performed at the 632.8 nm excitation support this interpretation (cf. Appendix A).

Most of the findings concerning the different a–C systems extracted from the Raman spectra are related to the G and D bands. These bands have a large width (≈200 cm^−1^) in all of the samples, being a characteristic of disordered systems [34]. The ratio of the intensities of the D and G bands, *I_D_*/*I_G_*, and the position of the G band are plotted as a function of the relative H + D content in Figure 3c. The increase of the H + D content in the films reduces the ID/IG intensity ratio from 0.75 to 0.45 (a similar trend has been reported in [35] due to the increase of the hydrogen content), while the G band position *σ_D_* shifts from 1563 to 1540 cm^−1^. While the observed trends are ambiguous when considered separately, a unique interpretation of this joint information is that the increase of the deuterium content corresponds to stage two of the well-known amorphisation trajectory of graphite, as proposed by Ferrari and Robertson [16]: the transformation of nano-graphite into amorphous carbon. In this amorphisation stage, the *I_D_*/*I_G_* ratio is directly proportional to the average number of carbon atoms in the graphitic clusters, i.e., to the square root of their diameter.

The fitting approach shown in Figure 3b was not applicable to the sample 10D due to the intense curvilinear background, mostly caused by the photoluminescence signal, which makes the extraction of information in the range of the D and G bands unreliable. Nevertheless, such an attempt has been made by removing the background of a polynomial form, which resulted in an asymmetric structure that could be mainly interpreted as the G band, while the D band is apparently lacking. Although of limited reliability, this result implies that the graphitic phase in the sample 10D is extremely small or even absent (see above). Details concerning the analysis of the Raman spectra of the sample 10D can be found in Appendix A.

In addition, an attempt was made to identify the character of the C-D and C-H bonds using other vibrational spectroscopies, namely FTIR and HREELS. These results are presented in Appendix A and Appendix B, respectively. In the case of the FTIR, the signal attributed to C-D bonds was identified only in the case of the sample 10D, revealing that about 60% of the deuterium atoms are bonded to sp^2^ carbon. A similar conclusion was obtained from the measurement of the sample 1D through HREELS, which confirmed that the character of the C-D bonds, being practically the same, is not responsible for the large difference between the electronic structures of the two samples. At the same time, the majority of the hydrogen at the surface of the sample 1D is bonded with sp^3^ carbon, clearly indicating that it corresponds to surface contamination caused by saturated hydrocarbons.

## 3. Discussion

The XPS results and the mismatch between the Tauc plot measurements and UPS spectra evidence that the samples are generally non-uniform. Hence, the films should be considered as mixtures of different carbon phases (being a frequent situation [16]), each characterised by their own local energy gaps. The steady increase of the films’ density with the increasing amount of incorporated deuterium up to the sample 1D can be understood as an augmentation of the amount of sp^3^ phases that are present in the film (cf. Table 1). At the same time, the sample 10D has the lowest density; despite the similar relative composition compared to the sample 1D, this aspect clearly indicates the formation of a polymeric phase [19]. These estimations are fully supported by the results of the XPS, UPS and Raman analyses, revealing that an increase of the deuterium content is accompanied by a decrease of the overall relative amount of the graphitic regions and their size reduction (i.e., transition from nanocrystalline graphitic phase into amorphous carbon). This is particularly evident when comparing the samples 1D and 10D, which are characterised by similar relative contents of deuterium (Table 1) and practically the same character of C-D bonds. The abrupt increase of the SEY and the Tauc gap from the sample 1D to the sample 10D can be only explained by the dramatic diminishing of the graphitic component (Table 1).

A similar conclusion was obtained by Robertson [18], who demonstrated a nearly linear dependence of the Tauc gap with the sp^2^ fraction *c_sp2_* for its relative contents below 80%, indicating that the relative hydrogen content does not have a direct influence on the variation of the band gap. As the data presented in [18] are compiled from the works of different authors and measured on various a–C and a–C:H samples, the points are rather scattered. Nevertheless, the line *E_T_* = 3–2.5∙*c_sp2_* represents well the observed trend with the Tauc gap uncertainty bar of about *ΔE_T_* = ±0.25 eV [36]. The dependences of the Tauc gap and the *SEY_max_* on the graphitic content obtained from the XPS analysis of our samples are shown in Figure 4. Both the *E_T_* and *SEY_max_* decrease when the relative amount of graphitic regions increases. Moreover, the trend of the Tauc plot fits well with that of Robertson—represented by a dashed blue line—for graphitic contents below 70%. The systematic shift between the two trends, which is within the uncertainty *ΔE_T_*, can be explained by the difference in the ways the graphitic content was determined [36] (using electron energy loss spectroscopy and nuclear magnetic resonance). In addition, at *c_sp2_* > 70%, we observe a faster drop of *E_T_*, as expected concerning the non-existent band gap of graphite. These agreements imply that the methodology used to extract the graphitic fraction from the XPS spectra is quite reliable and can also be used to correlate the SEY with the graphitic content.

The decrease of the SEY with the increase of the graphitic content (observed in Figure 4) is in accordance with the expectations: the higher the fraction of the sp^2^ phase, the lower the SEY. A simple model, based on the same assumptions as those used to derive the well-known semi-empirical equation for the SEY [37], was developed to support the observed experimental dependence. This equation, originally derived for uniform samples, was recently modified to encompass multilayer systems [38]. The latter approach was used as a starting point to model non-uniform a-C films as mixtures of two different phases: a graphitic one and a polymeric one. Therefore, the polymeric phase is considered to be representative of all non-graphitic phases in the films.

In the frame of the semi-empirical theory, the number of internal secondary electrons generated in a depth range of (*z*, *z + dz*) is equal to *S*(*z*)∙*dz*/*ε*(*z*), where *S* and *ε* represent the stopping power of the primary electrons and the effective energy required to create one internal secondary electron, respectively. The secondary electrons created at depth z will then be emitted with a probability equal to 0.5∙exp(−*z*/*λ*) [39], where *λ* is the mean escape depth of secondary electrons, which was taken from the literature [11]. As *S*, *λ* and *ε* are material dependent, they will be changing from point to point of the sample interior.

The first step is to determine the parameters that characterise the electron emission properties of the graphitic and polymeric regions. For this purpose, the SEY dependencies on the primary electron energy of the reference sample and the sample 10D [15] were used as representatives of pure graphitic and pure polymeric material, respectively. Then, for a defined relative graphitic content, it is possible to estimate the range of primary electrons *R*. The samples were then modelled as a multilayer system with a predefined graphitic content. However, the depth distribution of the different phases within each sample is unknown. Another difficulty in establishing this distribution is in the lateral non-uniformity of the samples. To overcome this problem, a set of *M* depth distributions were generated (here designated as configurations), each containing the same amount of graphitic carbon, using a Monte Carlo approach. Finally, the *SEY*(*E*) was calculated for each configuration by generating *S*(*E*,*z*)∙*dz*/*ε*(*z*) electrons at all depths *z* in the range [0, *R*], and calculating their escape probabilities affected by the multilayer structure in the depth interval [0, *z*]. The whole procedure is described in more detail in Appendix C.

The results of the SEY model, obtained using *M* = 1000 configurations and for the primary electron energies in the range 50–1000 eV are presented in Figure 4. Bearing in mind the simplifications introduced in the frame of the model, and that the calculated SEY curves were obtained without any adjustments to the existing experimental data, the agreement with the measured values is very good. There is also a built-in systematic error in the model due to the consideration that the reference sample would be purely graphitic, while its actual graphitic content is only 90%. This approximation is responsible for an overestimation of the SEY for the highest graphitic contents. Although the effective energy *ε* is indeed smaller in the polymeric (40.8 eV) than in the graphitic (60.4 eV) regions, the major difference between the characteristics of the two regions is the mean escape depth *λ*, being more than two times smaller in graphite (4.9 nm) than in polymers (11 nm). That is probably the reason the SEY_max_ dependence on the sp^2^ content has a form of an exponential decay function.

In a sample with a sufficiently high content of the graphitic phase, the regions with a large energy gap (e.g., diamond-like or hydrocarbon regions) will not contribute to the Tauc gap: photons that may penetrate large energy gap regions will most likely be absorbed once they enter a sufficiently large graphitic region. The reduction of the amount and size of the graphitic phase increases the optical transparency of the films and, consequently, the Tauc gap. A similar effect takes place during the transit of secondary electrons through the films. The escape depth of internal secondary electrons, which is directly related to their energy loss mechanisms, is strongly affected by the electronic structure of a material. The efficient energy loss is secured by sufficiently long trajectories of the secondary electrons through graphitic regions characterised by a very narrow energy gap. As in the case of light transmission, the graphitic phase serves as an energy absorber. Once they reach the surface, the secondary electrons will not have enough energy to overcome the energy barrier and cannot be emitted to vacuum. The increase of the graphitic content directly reduces the escape depth of the internal secondary electrons and, therefore, the secondary electron yield. While being supported by the presented experimental findings, this interpretation is rather simplified. Indeed, the *SEY_max_* of pure graphite is 1.3 [40], suggesting that imperfections of the graphitic phase in a-C might contribute to its low SEY.

Although the presence of hydrogen and deuterium in the thin films is not directly responsible for the SEY increase, it plays a key role in reducing the graphitic amount. Indeed, the H/D incorporation will naturally reduce the relative contents of both graphitic and diamond-like phases in a-C:H(D). It was revealed in our previous work that incorporation of deuterium is a consequence of target poisoning, yielding in the sputtering of CD and CD_2_ molecules in parallel with C atoms [15]. At the same time, pure carbon phases can only be deposited by magnetron sputtering of C atoms. The relative amount of pure carbon phases is directly related to the flux ratio of C and CD_x_ particles leaving the target and travelling towards the substrate. In the case of the sample 10D, this ratio was very low due to the strong target poisoning, resulting in a polymer-like film.

## 4. Materials and Methods

All details of the thin film deposition procedure are explained in [15]. Briefly, the films were deposited by magnetron sputtering from a 50 mm diameter graphite target. The total operating pressure was set to 2 Pa, consisting of Ar and small quantities of D_2_, to study deuterium incorporation into the films. The substrates were mounted 93 mm away from the graphite target and the discharge power fixed at 30 W. Five sets of samples were deposited on Si single crystal and quartz substrates for different characterisations: the reference (without D_2_ added to the discharge gas), 0.2D (0.2 vol% of added D_2_), 0.5D (0.5 vol% of D_2_), 1D (1.3 vol% of D_2_) and 10D (10.9 vol% of D_2_). D_2_ was added to distinguish the deliberately introduced contaminants from hydrogen contamination from the residual gas. The system was baked prior to each run for 24 h at 230 °C to minimise the natural contamination, keeping the residual gas pressure in the low 10^−6^ Pa range or better (N_2_ equivalent). A new graphite target was used for each run. The deposition rate was 10–15 nm/h, generally increasing for higher D_2_ partial pressure, p_D2_. The thin film thicknesses ranged between 486 and 719 nm (measured using cross-sectional SEM). The samples were transferred to different laboratories in stainless steel vacuum chambers, pre-evacuated using a turbomolecular pump and filled with N_2_ gas.

UPS and XPS measurements of ‘as-received’ samples were performed on an AXIS SUPRA setup (Kratos Analytical), containing both UV and monochromatic X-ray sources. XPS measurements were performed using the monochromated Al Kα line (photon energy of 1486.7 eV), and a spectrometer pass energy of 80 eV (survey spectra) and 5 eV (high resolution spectra). UPS measurements were performed by means of the He I α line (photon energy of 21.22 eV) and the spectrometer pass energy of 5 eV. The binding energy scale for the UPS and XPS measurements were both calibrated using a sputter-cleaned Ag sample, based on the position of the Ag 3d_5/2_ line and of the top of the valence band. Fitting of graphitic contributions was based on the XPS measurements of freshly cleaved Highly Oriented Pyrolytic Graphite (HOPG). A correction of the raw UPS spectra was made by removing the contributions of the He I β line (energy shift of 1.87 eV, intensity of 1.2% with respect to the He I α line). All samples were conductive, apart from the sample 10D, which was measured with charge compensation using an electron flood gun.

Raman spectroscopy measurements were carried out at room temperature in a backscattering geometry on an alpha 300 R confocal Raman microscope (WITec) using a 532 nm Nd:YAG laser (2nd harmonic), as well as a 633 nm He-Ne laser for excitation. The laser beam with a power of 0.7 mW was focused on the sample using a ×50 lens (Zeiss), providing a spot with a diameter of about 1 μm. The spectra were collected with a 600 groove/mm grating using 5 acquisitions with 2 s acquisition time. The same setup was used to perform photoluminescence measurements with two excitation wavelengths: 532 nm and 633 nm. For each sample, the spectra were acquired on several spots to check their lateral uniformity.

FTIR measurements were performed in a vacuum in the wave number range 500–5000 cm^−1^, using a Vertex 80v system (Bruker) in conventional reflection and transmission mode, as well as in the attenuated total reflection (ATR) configuration using a Ge crystal.

HREELS measurements were performed in a UHV system (operating pressure in the 10^−7^ Pa range) using a LK Technologies 2000R spectrometer. Further details on this experimental setup are provided in Appendix B, together with the experimental results.

## 5. Conclusions

Amorphous carbon coatings, deliberately modified through the incorporation of deuterium during magnetron sputtering deposition, were thoroughly analysed using different electron and vibrational spectroscopies. Adding deuterium enabled us to differentiate its contribution from that of hydrogen—the latter being present as a natural contaminant from the residual gas during the deposition. These data were combined with previous composition and optical measurements [15] to reveal the role of H/D incorporation in the change of the secondary electron emission characteristics.

The vibrational spectroscopies demonstrated that the majority of the deuterium is bonded with sp^2^ carbon, while the surface hydrogen is mainly present in the form saturated hydrocarbons. Both XPS and UPS (in combination with the Tauc gap) measurements revealed the non-uniform character of the samples, which is common for a–C materials [16]. A peak model was established, enabling us to determine the relative amount of carbon in the graphitic phase and to correlate this quantity with the SEY and the Tauc gap. The increase of the deuterium amount in the films is accompanied by a decrease of the fraction of the graphitic component and the cluster size. Based on the results, we conclude that the high content of graphitic phases in the films is responsible for the low secondary electron emission, in the same manner as it is for a low Tauc gap [18]. A model was developed to predict the SEY of non-uniform samples, which successfully describes the obtained experimental results.

The formation of pure carbon phases (graphitic or diamond like) and hydrocarbon regions are competing processes. In our experiments, the latter was the result of target contamination by D atoms formed in the plasma, followed by the physical sputtering of CD_x_ molecules, representing the building blocks of hydrocarbon phases [15]. Therefore, the flux ratio of the CD_x_ and C particles approaching the substrate was detrimental in this case for the relative graphitic content in the films.

The clusters of graphitic phases within the films have a local energy gap, close to 0 eV, which reduces the escape depth of electrons travelling through it and, consequently, also reduces the secondary electron yield.

## Figures and Tables

**Figure 1 ijms-24-12999-f001:**
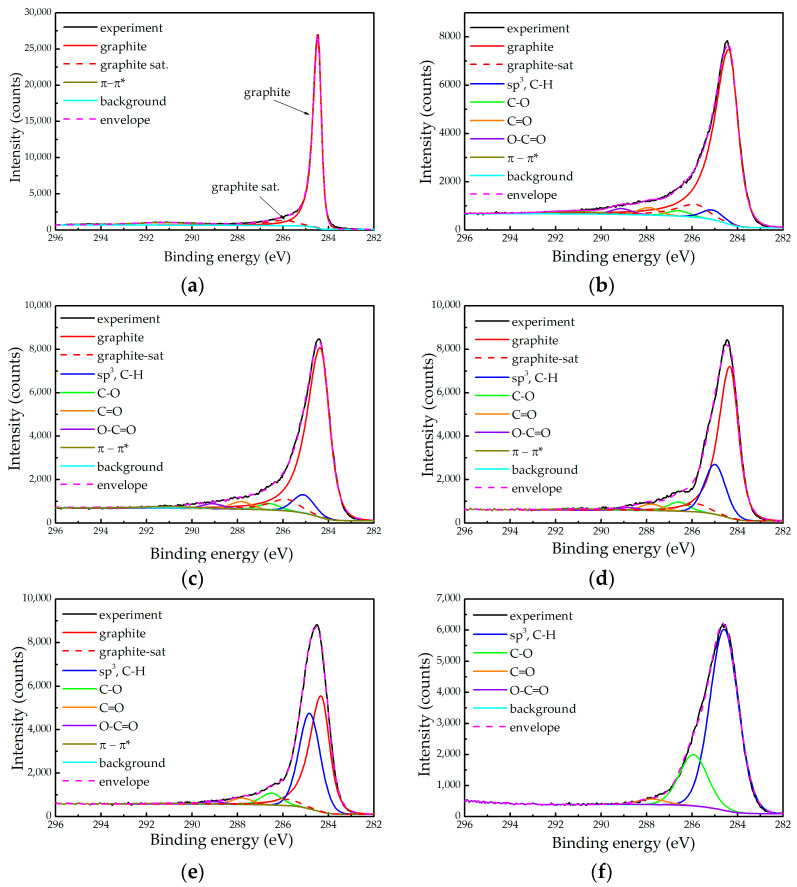
High resolution spectra of the C 1s state of (**a**) freshly cleaved HOPG; (**b**) reference a-C sample; (**c**) sample 0.2D; (**d**) sample 0.5D; (**e**) sample 1D; and (**f**) sample 10D, including the peaks obtained through data fitting, as described in the text.

**Figure 2 ijms-24-12999-f002:**
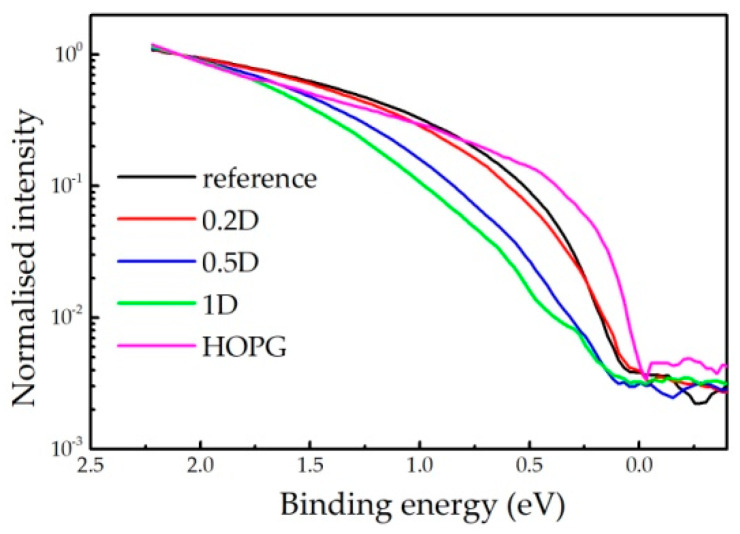
Valence bands electron spectra excited by He I radiation of HOPG and a-C coatings with different D content. The data are normalised at binding energy of 2 eV.

**Figure 3 ijms-24-12999-f003:**
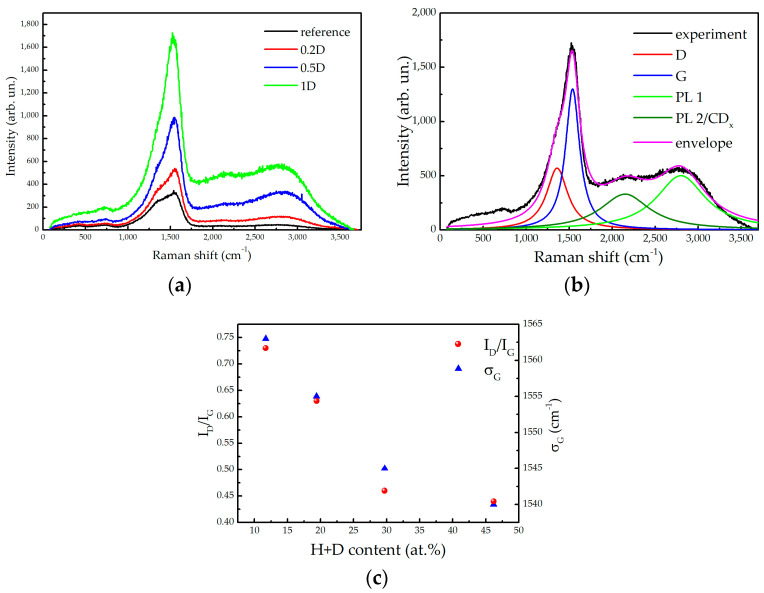
(**a**) Raman spectra of the reference sample, as well as of the 0.2D, 0.5D and 1D films deposited on Si substrates measured using a 0.7 W laser beam of 532 nm wavelength; (**b**) Fitting illustration on the example of the sample 1D; (**c**) intensity ratio of the D and G bands *I_D_*/*I_G_* and position of the G band maximum *σ_D_* vs. the relative H + D film content.

**Figure 4 ijms-24-12999-f004:**
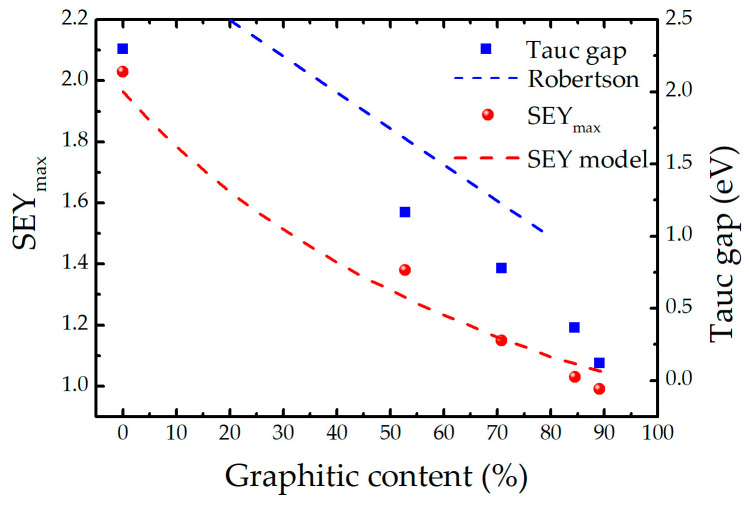
Dependence of the Tauc gap and the SEYmax on the graphitic content in a-C films. The dashed blue line represents the linear dependence of the Tauc gap as reported by Robertson [37]. The dashed red line is the prediction of the modified semi-empirical model for the SEY developed in this work.

**Table 1 ijms-24-12999-t001:** Variation of the composition (in at.%), *SEY_max_* and the Tauc gap and mass density of a-C films deposited by intentionally adding different amounts of D_2_ in the discharge gas [15].

Sample	p_D2_ (Pa)	C (%)	H (%)	D (%)	O (%)	SEY_max_	E_T_ (eV)	ρ (g/cm^3^)
Reference	0	83.4	11.7	0	4.9	0.99	0.12	1.44
0.2D	3.8 × 10^−3^	77.9	9.8	9.6	2.7	1.03	0.37	1.99
0.5D	1.1 × 10^−2^	67.8	6.0	23.7	2.5	1.15	0.78	1.91
1D	2.6 × 10^−2^	52.2	6.3	39.9	1.6	1.38	1.17	2.34
10D	2.2 × 10^−1^	45.2	2.8	50.4	1.6	2.03	2.29	1.15

**Table 2 ijms-24-12999-t002:** Summary of the relative amounts of different carbon bonds based on the C 1s line fittings in comparison with the relative H + D contents [15].

Sample	α/β	Graphitic (%)	sp^3^ C-H (%)	C-O (%)	C=O (%)	-(C=O)-O- (%)	H + D Content (%)
Reference	0.45	88.14	3.29	1.91	2.55	2.01	11.7
0.2D	0.50	85.17	6.43	2.41	2.57	1.73	19.4
0.5D	0.58	71.09	20.4	3.75	2.46	1.05	29.7
1D	0.58	52.75	38.11	4.83	2.46	0.96	46.2
10D	1.00	0	75.99	21.53	2.35	0.13	53.2

## Data Availability

Not applicable.

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
