# Peer review of "The Role of Hydrogen Incorporation into Amorphous Carbon Films in the Change of the Secondary Electron Yield"

_ijms, 2023, doi:10.3390/ijms241612999_

Round 1

Author Response

Response to Reviewer 1 Comments

Excellent work, it fully meets all the magazine's criteria,

- Correctness of assumptions

- Appropriateness of the experiments but also analysis of the results

- Detailed description of the experimental procedures

We thank the reviewer for the positive and encouraging comments.

You should pay attention to the following points:

-Amorphous carbon coatings, deliberately modified by deuterium incorporation during the magnetron sputtering deposition, were thoroughly analyzed using different electron and vibrational spectroscopies.  What if deuterium is not used?

Response 1:

The main motivation behind the usage of deuterium was to separate its contribution from the unavoidable natural hydrogen contamination from the residual gas during the deposition and the surface contamination mainly by hydrocarbons. The effect of deuterium on the electron structure of the samples (which governs the secondary electron emission) is expected to be the same as that of hydrogen. HREELS reveal different bonding character of the two species at the surface: C-H bonds are dominantly with sp3 carbon, corresponding to the surface contamination with saturated hydrocarbons, while D is more bonded with sp2 carbon atoms. This was already stated explicitly within the appendix A, and now it is also stressed at the end of the Results section and in the conclusion.

 - You should pay attention to English terminology

Response2 :

English has been revised

- The authors should state the novelty of their work more precisely in the conclusion part

Response3:

We thank the reviewer for this remark. The main novelties of the work are now stated explicitly in the conclusion.

Reviewer 2 Report

The presented manuscript is written thoroughly using good English. Unfortunately, the lines in the file are not numbered; therefore, I cannot mention concrete lines to recommend detailed improvements for the authors. Meanwhile, the authors should replace full references throughout the text with corresponding numbers. Moreover, I recommend highlighting some chemical outcomes of the performed study in the Results, Discussion, and Conclusions. After these corrections, the manuscript may be accepted for publishing in IJMS.

Author Response

Response to Reviewer 2 comments

The presented manuscript is written thoroughly using good English. Unfortunately, the lines in the file are not numbered; therefore, I cannot mention concrete lines to recommend detailed improvements for the authors. Meanwhile, the authors should replace full references throughout the text with corresponding numbers.

Response 1:

We apologize to the reviewer for this inconvenience. A problem occurred during the conversion of the Word file into pdf format. The problem is solved now. Concerning the language details, we passed through text and made some corrections.

Moreover, I recommend highlighting some chemical outcomes of the performed study in the Results, Discussion, and Conclusions. After these corrections, the manuscript may be accepted for publishing in IJMS.

Response 2:

The investigation reveals that the relative amount of graphitic character has the major effect on the secondary electron emission. In addition, in the revised version we further stress in different sections the character of chemical bonds between hydrogen and deuterium with carbon while the character of oxygen bonds with C is already detailed in Table 2. 

Reviewer 3 Report

The authors conducted an interesting study on amorphous carbon thin films with low secondary electron yield (SEY) to tackle electron multipacting in particle accelerators and RF devices. This article analyzed films produced through magnetron sputtering with varying hydrogen and deuterium contents, using XPS, Raman scattering, and UPS. They found that SEY decreases linearly with the graphitic phase fraction in the films. Additionally, the study demonstrated that SEY is influenced primarily by the graphitic phase rather than just the H/D concentration. The film's graphitic content reduces SEY due to its low energy gap and shorter electron escape depth. The writing style is easy and enjoyable, facilitating easy comprehension of scientific discussion. The authors' scientific approach and clear presentation make the article informative and easily understandable, even for researchers new to the subject. 

To enhance the manuscript further, the authors are recommended to address the following minor comments.

# How and why does the SEY variation depend on the graphitic phase apart from H/D concentrations? Need to explain clearly. 

# The current reference citation style with whole references within the text can be difficult to navigate. To improve readability and adherence to the journal guidelines, it is suggested to follow the appropriate reference citation format.

Author Response

Response to Reviewer 3 comments

The authors conducted an interesting study on amorphous carbon thin films with low secondary electron yield (SEY) to tackle electron multipacting in particle accelerators and RF devices. This article analyzed films produced through magnetron sputtering with varying hydrogen and deuterium contents, using XPS, Raman scattering, and UPS. They found that SEY decreases linearly with the graphitic phase fraction in the films. Additionally, the study demonstrated that SEY is influenced primarily by the graphitic phase rather than just the H/D concentration. The film's graphitic content reduces SEY due to its low energy gap and shorter electron escape depth. The writing style is easy and enjoyable, facilitating easy comprehension of scientific discussion. The authors' scientific approach and clear presentation make the article informative and easily understandable, even for researchers new to the subject.

We thank the reviewer for the highly affirmative comments. 

To enhance the manuscript further, the authors are recommended to address the following minor comments.

# How and why does the SEY variation depend on the graphitic phase apart from H/D concentrations? Need to explain clearly.

Response 1:

The main cause for the SEY reduction is the following: internal secondary electrons pass through graphitic regions where they survive energy losses and eventually do not have enough energy to overcome the energy barrier once they arrive to the surface. Therefore they cannot be emitted. In the 2nd paragraph from the bottom of the discussion (original manuscript) we state:

“The escape depth of internal secondary electrons, which is directly related with their energy loss mechanisms, is strongly affected by the electronic structure of a material. The efficient energy loss is secured by sufficiently long trajectories of the secondary electrons through graphitic regions characterised by very narrow energy gap. As in the case of light transmission, the graphitic phase serves as energy absorber.”

Now we add “Once they reach the surface, the secondary electrons will not have enough energy to overcome the energy barrier and cannot be emitted into vacuum.”, before the concluding sentence: “The increase of the graphitic content directly reduces the escape depth of internal secondary electrons and, therefore, the secondary electron yield.”    

# The current reference citation style with whole references within the text can be difficult to navigate. To improve readability and adherence to the journal guidelines, it is suggested to follow the appropriate reference citation format.

Response 2:

We apologize for this inconvenience that occurred during the conversion of the Word file into pdf format. The problem is now solved. 

Reviewer 4 Report

The manuscript describes experimental results on deposition and characterization of deuterated amorphous carbon films. Combination of experimental methods including light absorption UV and X-ray electron spectroscopy, Rama and IR spectroscopies, and other are used to reveal correlation between concentration of hydrogen/deuterium during the deposition and electronic properties of obtained materials.

Despite on low novelty of obtained results and triviality of their discussion the paper may be interesting for readers due to instructive and tutorial reasons.

However, to be suitable for that significant revisions are necessary. Authors should explain clearly why deuterium is used instead of pure hydrogen.

Before further evaluations full text description of the references must be removed from the text and replaced by numerations accordingly to common rules.

The manuscript must be checked for accordance of the general rules for academic publications including reference indication, minimization of empty space in figures, suitable editing of the figures and table legends (e.g. correct it for Table 1). 

Author Response

Response to Reviewer 4 comments

The manuscript describes experimental results on deposition and characterization of deuterated amorphous carbon films. Combination of experimental methods including light absorption UV and X-ray electron spectroscopy, Rama and IR spectroscopies, and other are used to reveal correlation between concentration of hydrogen/deuterium during the deposition and electronic properties of obtained materials.

Despite on low novelty of obtained results and triviality of their discussion the paper may be interesting for readers due to instructive and tutorial reasons.

Response 1:

We strongly disagree with the reviewer’s statement:

- An original peak model was built to resolve the relative contribution of graphitic carbon from the C 1s photoelectron line, and the Tauc gap vs. graphitic content result fits very well with those obtained using other experimental techniques (NMR and EELS), as can be seen in Fig. 4.

- For the first time, the dependence of the SEY on the graphitic content in a-C was presented 

- An original model was developed (in the frame of our “trivial discussion”) that tackles for the first time SEY of non-uniform materials, practically without fitting parameters, which successfully interprets the experimental findings (Fig. 4)…

Finally, there is nothing trivial in the result that the increase of the graphitic content up to 88% reduces the SEY of a-C samples, knowing that a-C has actually a lower SEYmax than pure graphite. Indeed, we omitted to mention this important detail in the original manuscript, which is now stated in the Discussion. While this paradox will be a subject of another paper that we are preparing, for now we can only say that this work is just an important new piece of the puzzle, while the overall understanding of secondary electron emission from carbon materials is certainly not yet fully resolved.  

However, to be suitable for that significant revisions are necessary. Authors should explain clearly why deuterium is used instead of pure hydrogen.

Response 2:

We used the deuterium to resolve its contribution from the natural contamination that occurs from the residual gas and the surface contamination by hydrocarbons after the deposition. This issue was extensively discussed within the ref. 15 and now it is explicitly stated in the Introduction, and repeated in the Conclusion.

Before further evaluations full text description of the references must be removed from the text and replaced by numerations accordingly to common rules.

Response 3:

We apologize for this inconvenience that occurred during the conversion of the Word file into pdf format. The problem is now solved. 

The manuscript must be checked for accordance of the general rules for academic publications including reference indication, minimization of empty space in figures, suitable editing of the figures and table legends (e.g. correct it for Table 1)

Response 4:

In the revised version, we have followed the formatting guidelines given by the journal and have paid attention to be as comprehensible in text, figures and captions as possible to enable for fluent reading.

Round 2

Reviewer 4 Report

The manuscript revision and replies to the comments from previous report are satisfactory.